# Effect of Animal Welfare on the Reproductive Performance of Extensive Pasture-Based Beef Cows in New Zealand

**DOI:** 10.3390/vetsci7040200

**Published:** 2020-12-11

**Authors:** Yolande Baby Kaurivi, Richard Laven, Tim Parkinson, Rebecca Hickson, Kevin Stafford

**Affiliations:** 1School of Veterinary Medicine, Massey University, Private Bag 11 222, Palmerston North 4442, New Zealand; r.laven@massey.ac.nz (R.L.); t.j.parkinson@massey.ac.nz (T.P.); 2School of Agriculture and Environmental Management, Massey University, Private Bag 11 222, Palmerston North 4442, New Zealand; R.Hickson@massey.ac.nz (R.H.); K.J.Stafford@massey.ac.nz (K.S.)

**Keywords:** beef cow welfare, extensive systems, reproductive performance, New Zealand

## Abstract

One key area where animal welfare may relate to productivity is through reproductive performance. This study assesses welfare on 25 extensively managed pastoral New Zealand beef farms, and explores the relationship between welfare and reproductive performance. Relationships between welfare measures and key reproductive performance indicators (pregnancy rate, weaning rate, mating period and bull: cow ratio) are investigated using an exploratory principal components analysis and linear regression model. Seven welfare measures (thinness, poor rumen fill, dirtiness, blindness, mortality, health checks of pregnant cows and yarding frequency/year) showed a potential influence on reproductive performance, and lameness was retained individually as a potential measure. Mean pregnancy rates, in both 2018 (PD18) and 2017 (PD17), were ~91% and mean weaning rate was 84%. Of the welfare measures, only lameness had a direct association with pregnancy rate, as well as a confounding effect on the association between mating period and pregnancy rate. The bull: cow ration (mean 1:31) and reproductive conditions (dystocia, abortion, vaginal prolapse) did not influence pregnancy and weaning rates. In the study population, there was no clear association between welfare and reproductive performance, except for the confounding effects of lameness.

## 1. Introduction

Beef productivity has generally been regarded as the outcome of genetic selection for production traits, such as growth rate and maternal milk yield, and management of nutrition and reproduction. Reproductive efficiency is a critical component of the overall performance of cow-and-calf beef cattle enterprises, for which the key performance indicators (KPI) of reproductive efficiency are pregnancy rate, calving and weaning rates, and calving to conception interval [1,2,3]. Many of the factors that affect these KPIs have been extensively studied, including, for example, macro- and micro-nutrition, changes in cow bodyweight, the effects of topography and climate, the presence of disease, and the bull to cow ratio [1,4,5]. Hence, indicators of good animal welfare, such as low incidences of disease, excellent nutrition, environmental comfort and good stockpersonship are all expected to give a direct or indirect association with reproductive performance of cattle, including beef cows in extensive production systems [6]. 

In New Zealand, there is an emphasis on improving reproductive performance in beef cattle. For example, the target is to have 95% of cows in the herd calving and 90% weaning a calf [7]. The proportion of herds achieving this KPI is relatively low [8], so a significant number of studies have examined the factors which allow beef cows to achieve this level of reproductive performance (e.g., References [8,9]). Although there is no standard recipe for achieving good reproductive performance, most New Zealand beef farms have adopted reproduction management plans, with monitoring of key monitoring KPIs, especially pregnancy rates, calving rates and dates, and managing parturition with assistance when necessary. In addition, reproduction management allows for the nutritional management of peri-parturient cows and adding value to planning for weaning and selling [10].

The relationship between the welfare of the cows and their reproductive performance has been described as complex, due to the limitation of predictive reproductive indicators for welfare and vice versa [11]. For example, good welfare alone does not necessarily lead to good reproduction, i.e., poor insemination techniques, poor bull performance and genetics, all can compromise conception [6]. Hence, reproduction measures were not fully supported as direct measures of animal welfare [11]. However, reproduction can have direct adverse effects upon animal welfare through, for example, catastrophes, such as dystocia, abortion and metritis. Poor performance in other reproductive KPIs, such as low conception rates and long post-partum anoestrus, particularly where many cattle are affected, may also provide an indication that there is poor herd welfare [12]. The relationship between the welfare of the cows and their reproductive performance has received relatively less attention than nutritional factors, but, more recently, evidence has started to accumulate that both production and reproduction are influenced by welfare [13,14,15,16]. There is strong evidence from other domestic species, such as rabbit does [17] and sows [18,19], that welfare is a significant factor for determining reproductive performance. Whilst it has not been clearly established whether, or to what extent, suboptimal animal welfare is related to suboptimal reproductive KPIs, recent studies found a direct correlation between welfare measures and reproductive performance of confined beef [15] and dairy [16] cows in different housing systems. 

Hence, the current paper aims to evaluate whether there is a relationship between several animal welfare indicators and the reproductive efficiency parameters of beef cows in New Zealand managed in extensive pasture-based systems. It was postulated that farms with poor welfare indicator scores would be correlated with poor reproductive KPIs. The aim was achieved through a series of factor analysing and selecting the likely measures to predict welfare and reproductive performance. Results from this study are foreseen to fill a gap in the knowledge of how animal welfare quality holistically affects the reproductive performance of beef cattle that are extensively reared.

## 2. Materials and Methods

### 2.1. Welfare Assessment and Reproductive Data Collection

The welfare of beef herds was assessed using the Welfare Assessment Protocol (WAP) developed by Kaurivi [20,21]. The WAP was developed as an amalgamation of the best practicable measures in the Welfare Quality protocol [22] and the rangeland-based University of California Davis (UC Davis) Cow-Calf Health and Handling assessment protocol [23], with additional measures suitable for New Zealand beef systems.

This was an observational study following veterinarians on their routine pregnancy testing. Cows were not handled or yarded for a longer duration because of the study, had no additional induced activity/yarding/handling, and the presence of the observers did not influence their behaviour. Thus, ethical approval was waived by the local ethic committee.

The study was carried out on 25 extensive pasture-based beef herds in Waikato district of the North Island of New Zealand. This was a convenience sample of the beef cattle clients of a local rural practice, who used the practice for routine pregnancy diagnosis in autumn. Breeding cows were extensively reared on pasture mostly on a hill or high country, complimentarily with sheep. All farms used rotational grazing as their main means of feeding cattle, with supplementary hay or silage given during winter when necessary. No housing or off-pasture feeding was used on any farm. Average herd size (cows mated) was 198 (range 18–541). The predominant cow breeds were Angus and Hereford and their crosses, as well as with dairy crosses. The water was generally sourced naturally and abundant or provided in troughs in paddocks (the average amount of annual precipitation in the area is 1119.0 mm). The average annual minimum temperature is 9.0 °C, and maximum temperature is 19.0 °C [24]. Subjective assessment of shade in the paddocks (presence of trees, shrubs, galleys) showed that this was sufficient on all farms except one.

Each farm was visited twice. The first visit took place in the autumn (March/April 2018) at the time at which pregnancy diagnosis was undertaken. During the first visit, more than half of the animals in each herd were directly observed against the parameters of the WAP whilst they were yarded for pregnancy diagnoses. The second visit was a few months later, in winter, at a time when the animals were not yarded. This visit involved a farm resource evaluation and a questionnaire-guided interview to assess the health and management of each herd in the last 12 months. The general cattle management and key health aspects on the farm (disbudding, castration, vaccination, diseases history, cattle deaths, access and type of water supply, feed/pasture condition, wintering practices) were recorded. 

Pregnancy rates for 2018 (PD18) were determined at the same time as the animal welfare assessment was undertaken in the yard, while similar data for 2017 (PD17) were retrieved from the veterinary clinic records. The questionnaire-guided interview captured data of cows’ reproductive conditions (retained fetal membranes, prolapse vagina/uterus, dystocia and abortion) and their potential causes and outcomes in the previous 12 months. Other reproductive data, including bull: cow ratios, start dates of mating and durations of mating periods, were summarised along with numbers of calves born, tagged and weaned from 2017 pregnancy testing. Calf mortality was calculated from calved weaned from in-calf cows. From these data, reproductive KPIs were calculated using an adaption of Hewitt [3]: Table 1. 

To interpret the collected data of animal welfare measures in a meaningful way, the measures were integrated into the four animal welfare principles (good feeding, appropriate environment, good health and appropriate stockpersonship: Table 2). For each farm, the welfare impact of each measure was categorised separately in three categories; 0: poor; 1: marginal and 2: poor/unacceptable welfare. All discrete data were measured according to the percentage of cases and given a 3-point ordinal welfare score. Descriptions of how each measure was assessed, and categorised, as reported by Kaurivi [21].

### 2.2. Data Analysis

All data were analysed using SPSS IBM version 27. Descriptive statistics for continuous measures were mean, median, standard deviation, range and percentiles. Qualitative methods were used to analyse the frequency of ordinal measures. The welfare data were correlated with the KPI of each herd. To determine how welfare predicts reproductive performance, principal components from 32 welfare measures were used as predictors for reproduction performance. Relationships between reproductive variables (PD17 and PD18 rates, weaning rate, mating period and bull: cow ratio, dystocia, abortion, vaginal prolapse) and the welfare measures were investigated using an exploratory principal component analysis (PCA). Seven variables were selected in the PCA; two were related to good feeding (thinness and poor rumen fill), one was related to the appropriate environment (dirtiness), and two were related to good health (blindness, mortality) and two to appropriate stockpersonship (yarding frequency/year, health checks of pregnant cows). Principal components were identified based on Eigenvalue > 1. After the extraction of the data, an Oblimin rotation was used to show a correlation [25]. An individual score for each component was assigned to each measure. Based on explanatory components, the seven measures for PCA were chosen based on: (1) Diversity across farms and (2) likely impact on reproductive performance. Lameness and dystocia were added individually as part of the measures predicting reproduction performance based on the likelihood of their effects.

Data were further analysed with a linear regression model. The component scores (BART factor scores) for the seven selected welfare measures along with lameness, bull: cow ratio and mating period were then used as predictor variables with one of the reproductive outcomes (PD18, PD17 and Weaning 17) as dependent variables in separate regression models. Backwards stepwise selection of variables was used (target *P*-value < 0.05).

## 3. Results

### 3.1. Reproductive Data

Basic reproductive performance data for the 25 herds are summarised in Table 3. Qualitative data for reproductive conditions (pooled data for abortions, dystocia and vaginal prolapse) are summarised in Table 4. The mean pregnancy rate for 2018 was 91.4%, with only three farms achieving pregnancy rates of <85%. The PD18 was like PD17 with 91.9% mean and the same median at 92.7%. The number of calves born was recorded at only 4/25 farms, tagging record was for 18/25 farms (at 2–4 months), while 3/25 farms tagged calves only at weaning (mode six months). Calving rate (93%) could only be calculated at the 4/25 farms that recorded the number of calves born. Mortality of calves from births to tagging (95.6%) was also only calculated at these four farms. Losses from tagging to weaning were calculated for 22 of the farms (excluded 3/25 farm that only tagged at weaning). Calf mortality varied from 0 to 9%, with an overall mean of 2.4%, and 10 (48%) farms achieving <2.5% losses. The mean weaning rate was 83.9%, with only six farms achieving ≥90%. The proportion of calves weaned from known pregnant cows was 92.3%. Bull: cow ratio was between 1:20 and 1:40 for all but two farms (one 1:12, the other 1:62). The mating period was between 42 and 72 days in 16 (64%) farms, four had short mating periods of 30–41 days, whilst two had mating periods of ≥100 days. There were relatively few reproductive disorders, with an overall average for all disorders in 2017 of 2.3% of the entire cohort of 4637 calving cows (Table 4). Vaginal prolapse was very rare (0.2% of calvings). Overall, 2.6% of calvings were assessed by the farm manager as dystocias, but these data are skewed by one farm that reported an incidence of 17%. All other farms had an incidence of <5.5%. Twelve farms reported no abortions, whilst two reported 9%. No cases of retained fetal membranes were reported at any of the farms.

### 3.2. Welfare Data

Descriptive statistics for welfare measures are shown in Table 5 and Table 6. For *good feeding*, the average of thin cows at 10.7% was skewed by one farm with 61%, whilst 50% of farms had ≤5.7% of thin cows. There was an average of 30.6% of cows with poor rumen fill. For appropriate environment, 21.3% of cows were assessed as being dirty, and 39.6% had faecal soiling. Dirtiness was correlated with the presence of faecal soiling (r = 0.75; [20]). For most health-related measures, the prevalence was low on most farms. The exceptions were lameness and mortality rate. There was an average of 2.7% lame cows (maximum: 12%). Mortality, as presented by accidental deaths, deaths due to diseases and culling for health issues at the herds, were present at an average of 3.9% of cattle. Blindness was rare at the farms with an average of 0.4% (0–4%). For stockpersonship, an average of 2.7% of cows displayed fearful/agitated behaviours (attempt to escape in the race, climbing/pushing others). Some farmers (11/25) did daily health checks on pregnant cows, whilst 9/25 inspected cows once or twice a week, and others 5/25 did not inspect within a week. For yarding frequency, no farms yarded cows two times or less per year, and most farms (20/25) were yarded between 3–4 times per year. For painful management procedures, castration was performed on 20 of the 25 farms (mode and median 2 months of age; range 1 to 4 months). Only two farms disbudded calves (at 3 and 4 months respectively). Ear tagging was performed at all farms with median and mode of 2 months.

### 3.3. Prediction of Welfare Effects on Reproduction Variables

See Appendix A for descriptive statistics of measures selected for inclusion in the prediction of welfare effects on reproductive performance for each of the 25 farms in Waikato. 

The seven welfare measures subjected to PCA, showed many coefficients of 0.3 and above. The Keiser-Meyer-Olkin (KMO) of 0.058 was an acceptable threshold, and Bartlett’s tests reached a statistical significance of <0.001. The PCA revealed the presence of three components with an eigenvalue of >1, explaining 73.9% of the variance with 15%, 19.9% and 39% of the variances, respectively. An inspection of the scree-plot revealed a break after Component 2, which was retained for further analysis. The highest extraction was for thin cows (0.929) and lowest for yarding frequency (0.265). The two components explained a cumulative variance of 58.9% with 39% in Component 1 and 19.9% in Component 2. Component 1 loaded for thin, poor rumen fill and blindness and Component 2 was for mortality, yarding frequency/year and dirtiness (Figure 1). The rotated solution revealed the presence of a simple structure with components showing loadings and all variables loading on only one component, except health checks that loaded on both. The interpretation of the components shows a relationship between variables in each specific component. There was a slight positive correlation between the two factors r = 0.151. The results of this analysis showed a strong association between health-related measures (blindness, lameness), but mortality showed a weaker association. Health checks and yarding showed an association, although yarding frequency associated more with the health and feeding measures. Dirtiness showed a negative association with health and feeding measures. There was a strong correlation between thin cows and cows with poor rumen fill (r = 0.714), as well as a strong correlation between thin cows and blindness (r = 0.892). PD17 and PD18 were negatively correlated (r = −0.217). 

Linear regression modelling showed:

PD18 dependent variable: Mating period and lameness were the variables remaining at the end of the selection process. At a constant mating period, the coefficient showed that an increase of 1% in lameness was associated with an increase of 0.89% in PD18% (95% CI). At a constant lameness rate, the increase of mating period by one day decreased PD18 by 0.24% (95% CI). A 1% increase in lameness was associated with four days longer mating period. Further analysis with Cooks Distance removed one influential farm that greatly affected the slope of the regression line. For PD18, lameness and mating period then became confounders.

PD17 dependent variable: Mating period was the only variable remaining at the end of the selection process. The coefficient indicated that removing lameness changed mating period from 0.13 to 0.08, a change of ~40% (95% CI), indicating that lameness was a confounder. So, a one day increase in mating period (before 2018 PDs) was associated with an increase of 1% in PD17. Having accounted for lameness prevalence, the effect of the mating period on PD17 was 1.4% (95% CI). Having accounted for the mating period, lameness reduced PD17 (but the data was compatible with a large negative impact −1.3 days per percent rise and a small positive benefit 0.24 for a 1% rise). There was no influential farm in PD17, and Cooks Distance was not analysed.

Weaning 17 dependent variable: Measures in Component 1 (thin and poor rumen fill cows, and blindness) were the variables remaining at the end of the selection process. For an increase of one unit (≡1 SD) in the component score, weaning increased up by 3.5%. The weaning rate was higher with an increase in these factors, but was negatively correlated with health checks.

## 4. Discussion

The current study postulated that animal welfare influences the reproductive performance of beef farms managed extensively on pasture. Of the many reproductive KPIs that can potentially be used to assess the performance of beef cattle, it was difficult to standardise these across the 25 farms in the present study because of incomplete record-keeping. The most consistently available KPIs were pregnancy rates, weaning rates, bull: cow ratio and duration of the mating period. Moreover, the reproductive KPIs that are relevant and assessable within extensive, pastoral beef cow-calf systems differ markedly from those that are feasible in-housed/feedlot systems [3]. Thus, in the latter systems, data, such as calving intervals, calf to cow ratio per year, number of inseminations per conception, mean pregnancy to cow ration, age at culling and cattle culling %, are readily available [15,16]; but none of these could be ascertained or was pertinent to bull-bred cows in extensive pastoral systems. Thus, direct comparison of reproductive performance and welfare effects with such studies is not particularly meaningful, even though their overall conclusion that good welfare standards are related to improved reproductive performance in cattle was not clearly reinforced by the present results. It has been argued that welfare data should be compiled into an ‘overall’ score [26], but, on the other hand, others [27,28] have rejected such compilation of scores as being complex and covering up important welfare problems. Rather, the influence of welfare on reproductive performance was assessed through analysis of individual welfare measures and not as an overall farm score as was done in Grimard [16]. The relationship between welfare and reproduction parameters in this study was assessed at a herd and not at an animal level.

Average pregnancy rates (PD17: 91.9% and PD18: 91.3%) were consistent with other studies in New Zealand (91%: [8], 90.8–91.0%: [5], but were below the target of 95% set by the beef industry [7]. In terms of relationships with welfare measures, the present study found no relationship between herd thinness and poor rumen fill and pregnancy rate. The BCS evaluation coincided with weaning, which may not necessarily reflect BCS at mating, which is more important for reproductive performance. Morris [29] finding that cows with higher BCS at mating have better pregnancy rates than thinner cows, and Weik [5] showing that ideal body weight (e.g., 7 on 1–10 scale) at mating was associated with an improvement of pregnancy rates. Furthermore, access to adequate feed and water is widely recognised as influencing both the health and welfare [30] and productivity and fertility of cows [29,31]. 

The current findings suggest that the nutritional status of herds in this study was not sufficiently compromised to have had a negative impact on fertility: As also suggested by Probert [32]; Pleasants and Barton [33], and Hickson [34]. Indeed, in the extensive pasture-based systems of beef cattle in New Zealand, nutritional problems are only sufficiently severe to affect the reproductive performance of cattle during drought periods [7,32]. On the other hand, cattle in New Zealand are prone to copper deficiency (which contributes to lowering fertility and weaning rate: [7,35], which may have confounded the relationship between BCS and fertility. Assessing the micro-nutrient status of the cows might help to resolve this conundrum (see References [36,37]). 

Lameness is a critical welfare compromise indicator, as it is both a painful condition and affects productivity and fertility [38,39]. There is an increasing amount of literature that suggests lameness negatively impacts the reproductive performance of cows. For example, various studies have shown that lame beef cows have an increased service period, increased calving to conception interval, increased number of services per conception, impaired follicular growth, ovulation and oestrous behaviour, and hence, less chance of getting pregnant [39,40,41,42]. In the present study, herd lameness was associated with a longer (4 days) mating period. It was also a confounder for the mating period when PD17 was the predictive variable, i.e., indicating that lameness was associated with reduced PD17. Paradoxically, however, farms with high levels of lameness and long mating periods had high PD18 and vice versa. These results show that lameness confounds the impact of long mating periods on overall pregnancy rate. This may be because on farms where lameness is not a problem—long mating periods indicate that there is limited focus on getting cows pregnant (and thus, low final pregnancy rates compared to farms with the same level of lameness and shorter mating periods). Whereas, on farms with high levels of lameness, long mating periods are used to increase the chances of cows becoming pregnant (and are, therefore, associated with higher final pregnancy rates than farms with the same level of lameness and shorter mating periods).

Other welfare measures were unrelated or only weakly related to pregnancy rate. Dirtiness was unrelated, probably because, in the present study, it was only an indicator of the lushness of the pasture [21,43]. This contrasts with the situation with housed cattle, where dirtiness is related to both risks of disease and of failure to conceive ([44]. Likewise, measures of stockpersonship, such as yarding frequency, cows’ behaviour in the yard were unrelated in the present study to any reproductive parameters. Interestingly, the frequency of yarding was related to the incidence of fearful behaviour (i.e., cattle attempting to escape the race or climbing/pushing on others), where cows that were yarded few times per year were more fearful, as previously recorded [45,46]. Others have shown that fearful/agitated behaviours in the yards could be related to temperamental cows (i.e., as measured by flight speed) and were associated with reduced fertility [14,47], possibly due to stress-induced increases in plasma cortisol concentrations [48], so the lack of an association in the present study was not expected. Accepting that temperament and flight zone were not included as part of the welfare assessment (see Kaurivi [20,21] for why), it appears that, in the present study, the commonness of fearful/agitated behaviour was just a temporary indication of the lack of familiarity of extensively managed cattle with yarding and handling, and not necessarily a permanent situation. 

The average weaning rate (proportion of weaned calves from mated cows) was 83.9%, which is below the industry target of 90% [7,49]. However, Reference [49] indicated a very similar figure (80–84%) in a survey of cow-calf herds in New Zealand. However, the proportion of calves weaned from recorded pregnant cows (i.e., rather than mated cows) was 92%. If the number of pregnant cows is used as a proxy for expected calves to be born, it indicates an effect of lower conception (~6% than calf survival to weaning). Thus, remedial strategies geared towards improving the conception rate might be more pertinent to increase reproductive performance in the national cow-calf herds than focusing on ensuring calf survival. There was, however, a paradoxical relationship between welfare measures and weaning rate in the present study, since the weaning rate was positively related to an increase in thinness, poor rumen fill and blindness in cows. Again, these unexpected results could be due to the low rate of problems in the study herds. Most other reports, e.g., References [50,51,52], show the opposite, i.e., that good nutrition and health are associated with improved pregnancy and weaning rates. Whether this discrepancy can be attributed to peri-partum or pre-weaning mortality is a moot point, but the present results may merely indicate that the mortality rate (average 3.9%) was not high enough to influence reproductive performance. 

The average duration of the mating period (62 days) was closely aligned with the industry standard of 63 days [49]. The target is to at least have 60% of cows calving in the first 21 days of calving to avoid spread-out calving and to indicate a high reproductive performance of a herd [7]. The mating period has also been shown to be related to pregnancy rates and the incidence of post-partum anoestrus [8]. Ideally, the interval from calving to conception would have also been measured, as it has been indicated that this could be an indicator of animal welfare [53], but this is not systematically recorded in extensive beef systems (as also found by McFadden [8]). On the other hand, there is evidence that calving to conception interval may not be a particularly useful welfare indicator, since even in dairy cows (in which this is routinely recorded), it is disregarded as a routine performance indicator for animal welfare [54]. 

The median bull: cow ratio was 1:30. Previous studies of reproduction management plans in New Zealand show that bulls in natural breeding systems can mate with 30–40 cows [7]. Whilst there was no relationship between bull: cow ratio in the present study, it was clear that farmers tended to reduce the number of cows per bull to reduce the risk of pregnancy failure in the herd (as also found by McFadden [8]. However, over the range of ratios of 1:25 to 1:50, the literature shows only a tenuous relationship between bull: cow ratio and pregnancy rate (e.g., Reference [55]). Thus, the 1:30 ratio observed in the present study is probably a reflection of current recommendations regarding maximising the dissemination of ‘superior’ genetics ([7]) through optimising bull: cow ratio. Farmers also reported to purchasing bulls certified from reputable breeders to limit the use of low performing bulls, although bulls were not routinely tested before mating.

The average prevalence of dystocia in the present study was 2.6%; half the industry standard of 5% [49]. Dystocia has a very significant impact upon the welfare of affected animals (and upon calves born/stillborn as a result of dystocia), and at high incidences, can markedly impair the productivity of the farm [34,56]. Hence, our previous study [21] recommended a threshold of >2% for dystocia as indicative of poor welfare that needs immediate intervention. However, the present study did not show a relationship between the prevalence of dystocia and reproduction outcomes. Interestingly, the prevalence of dystocia was unrelated to BCS, as also shown for pastoral beef cows by Hickson [57]. The prevalence of abortion was also low (1.5%), well below the 3% industry target [49] and was unrelated to any other parameters of reproductive performance. High incidences of abortion are, in extensive beef systems, usually related to diseases, such as BVD infection and leptospirosis [58]. No individual herd in the present study had a high prevalence of abortion, and only one herd in this study reported abortion losses, due to BVD, others were in individual cows, due to unknown causes.

This was a relatively small study in terms of the number of herds enrolled, although very extensive welfare data were gathered from each herd. However, the results suggest no clear impact of welfare at a whole-of-study level, and only an individual farm association between animal welfare measures and reproductive performance. This may be because most of the herds in this study did not have a welfare compromise as the mortality rates, and % of thin, lame cows or cows with health problems was very low. In addition, the lack of records limited the type of reproductive performance that could be assessed in this study. More data from more farms across New Zealand are required to better understand the association between fertility and welfare on pasture-based extensive beef farms in New Zealand. Further studies that consider confounding factors for reproductive performance, such as cow breed, parity, calving to conception interval and bull performance [16], may also alleviate the complexity in investigating the association of welfare and reproductive performance [11] in extensive beef cattle systems.

It was evident that variations may occur in farm reproductive performance across years, as was shown by the negatively correlated pregnancy rates in the two consecutive years in this study. In contrast, we collected animal welfare data at only two time points in one single year. Further studies should, thus, evaluate the relationship between welfare standards and reproductive performance over multiple years. The results of this study showed that the impact of poor welfare on fertility is not necessarily consistent; it depends on the system and the farm, and likely on individual animals. Whilst average reproductive output may be good on most beef farms in New Zealand, the welfare of some individual herds may be poor; consequently, alternative strategies should be made for the welfare improvement of these individuals [59]. Poor recording meant that many potential reproductive KPI could not be collected for this study, but the study reinforced that routine pregnancy testing is critical to identify infertile cows and heifers to make appropriate management decision, such as culling of poorly fertile cows. Hence, pregnancy rates are good indicators of cows’ reproductive performance [7]—and coupled to weaning rate, these factors do give an overall indication of the reproduction performance in beef herds.

## 5. Conclusions

This study investigated the relationship between animal welfare and the reproductive performance of extensive pasture-based beef cows in New Zealand. The use and analysis of a newly developed animal welfare assessment protocol [20,21] for New Zealand cow and calf systems showed that seven welfare measures (thinness, poor rumen fill, dirtiness blindness, mortality, yarding frequency/year and health checks frequency of pregnant cows) may potentially influence reproductive KPI. Lameness was also retained to the potential measures. The key findings were that feasible KPI across the cow-calf farms were pregnancy and weaning rates, which coincided with results from previous studies. From the welfare measures, it was only lameness that showed a direct effect upon pregnancy rate, as well as showing a confounding increase in pregnancy rates with an increase in the mating period. The bull: cow ratio and reproductive conditions (dystocia, abortion, and vaginal prolapse) did not influence pregnancy and weaning rates. Overall, the study showed no clear evidence of an effect of animal welfare standards on the reproductive performance at the herd level. Our results provide the first evidence for using measures from an animal welfare assessment protocol and associated welfare outcomes with reproductive KPI in extensive beef cow-calf operations. Further research on multiple beef farms is warranted to clarify these findings. The integrated approach on how some indicators of animal welfare might indirectly affect the reproductive performance of beef cattle in an extensive production system could be beneficial to address mitigation or intervention strategies for the improvement of reproductive efficiency in beef cattle. 

## Figures and Tables

**Figure 1 vetsci-07-00200-f001:**
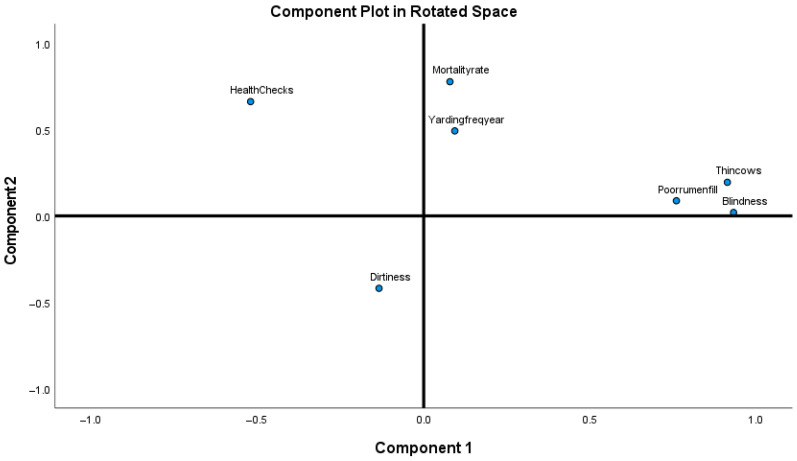
Principal components analysis (PCA): Components showing animal welfare measures rotation.

**Table 1 vetsci-07-00200-t001:** Calculation of reproductive key performance indicators. KPI, key performance indicators.

Reproductive KPI	Calculation	Raw Data
Pregnancy rate	(A−E)/A%	(A) Number of cows mated (B) Number of bulls (C) Date bulls turned in to herd (D) Date bulls removed from the herd (E) Number of non-pregnant cows (F) Number of calves born (G) Number of calves tagged (H) Number of calves weaned
Barren rate	E/A%	
Calving rate	F/A%	
Bull: cow ratio	B: A	
Mating period	D−C days	
Weaning rate (weaning/mated cows)	H/A%	
Weaned/known pregnant cows	H/Pregnant	
Calf mortality to tagging	(F−G)/F%	
Calf mortality tagging to weaning	(G−H)/F%	

**Table 2 vetsci-07-00200-t002:** Ascribing welfare measures into welfare principles.

Welfare Principle	Welfare Measures
Good feeding	Body condition, rumen fill, access to water
Appropriate environment	Dirtiness, short tail, hazards, shade, hazards
Good health	Swelling, abrasions, hair loss, blindness, ocular discharge, nasal discharges, diarrhoea, lameness, dystocia, mortality, castration, disbudding, ear tag/notching
Appropriate stockpersonship	Fearful/agitated behaviour, mis-catching, hitting, fall, run, stumble, health checks, yarding frequency, yard design

**Table 3 vetsci-07-00200-t003:** Summary of reproductive performance data of beef cows in 25 Waikato farms.

	% Pregnant 2018	% Pregnant 2017	% Weaned 2017	Bull: Cow Ratio	Mating Period (d)
Mean	91.4	91.9	83.9	1:31.3	61.8
Median	92.7	92.6	83.3	1:29.7	61.0
Range	69.2–97.9	81.4–100	67.8–96.7	1:12–1:62	30–125
Std dev	5.8	5.1	7.7	9.3	21.4

**Table 4 vetsci-07-00200-t004:** Summary of reproductive disorders experienced by beef cows (in-calf 2017 mating) in 25 Waikato farms.

	Herd Size	Vaginal Prolapse (n/farm)	Dystocia (n/farm)	Abortion (n/farm)	Total n/farm
Mean	185.5	0.2	2.6	1.5	4.3
Median	153.0	0.0	1.9	0.6	2.5
Proportion of entire calving cohort		0.1%	1.4%	0.8%	2.3%

**Table 5 vetsci-07-00200-t005:** Descriptive statistics (from 25 Waikato beef farms) for measures recorded as a percentage of observed animals. The description of assessments used to create these figures was reported by Kaurivi [21].

		Descriptive Statistics					
Animal Welfare Principles	Animal Welfare Measures	Min	Max	Mean	Percentiles		
					25	50	75
Good Feeding	* Thin cows	0	61	10.7	2.6	5.7	10.0
	* Poor rumen fill	0	68	30.6	15.5	29.9	45.7
Good Environment	Short tail	0	21	4.2	0.6	3.0	6.0
	* Dirtiness	4	50	21.3	10.7	20.6	29.4
	Diarrhoea (faecal staining)	15	87	39.6	24.0	35.7	48.5
Good Health	Swelling	0	5	0.7	0.0	0.0	1.1
	Hair loss	0	1	0.1	0.0	0.0	0.0
	Abrasion	0	2	0.1	0.0	0.0	0.0
	* Lameness	0	12	2.7	0.5	1.5	3.6
	* Blindness	0	4	0.4	0.0	0.0	0.0
	Ocular discharge	0	8	1.5	0.0	0.0	3.2
	Nasal discharge	0	13	1.2	0.0	0.0	1.3
	* Accidental deaths	0	2	0.6	0.0	0.4	1.2
	* Deaths from disease	0	7	2.0	0.9	1.6	2.9
	* Culling for health	0	6	1.2	0.0	0.8	2.0
Stockpersonship	Fearful/Agitate	0	7	2.7	1.3	2.3	4.1
	Fall/lie	0	8	0.9	0.0	0.0	0.8
	Stumble	0	21	1.6	0.0	0.0	1.7
	Run exit	0	51	13.0	2.6	7.8	15.1

* Measures selected as having high potential effects on reproductive KPI.

**Table 6 vetsci-07-00200-t006:** Observed frequencies for categorical stockpersonship measures at the 25 Waikato beef farms. Description of assessments used to create these figures were reported by Kaurivi [21].

Stockpersonship Measures	Categories and Number of Farms in Each Category		
Mis-catch	No mis-catch	˂1% of cows mis	>1% of cows
	18	4	3
Hitting	No hitting	Few cows hit	˃10% hit
	18	4	3
Noise of handlers	No noise	Minor audible noise	Noisy handlers
	4	18	3
Noise of Equipment	No noise	Minor audible noise	Very noisy equipment
	9	6	10
Dogs noise around the yard	No dogs around	Quiet dogs	Noisy dogs
	7	8	10
* Health checks (pregnant)	Daily inspection	Once or twice a week	Less than weekly
	11	9	5
* Yarding frequency	>4 times/year	3–4 times/year	Two times or less
	5	20	0
Yard (design) flow of cows	Effective	Minor problems	Significant problems
	13	7	5

***** Measures selected as having high potential effects on reproductive KPI.

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
