# Peer review of "Effect of Animal Welfare on the Reproductive Performance of Extensive Pasture-Based Beef Cows in New Zealand"

_vetsci, 2020, doi:10.3390/vetsci7040200_

Round 1
Reviewer 1 Report
Lines 220 (Discussion): based on the results of this work I would like to see in the from the authors discussion about the difficulty of apply reproduction parameters as direct or even indirect indicators of cattle welfare. As suggestion see below some paper about this subject:
Fraser, D., Duncan, I. J., Edwards, S. A., Grandin, T., Gregory, N. G., Guyonnet, V., … Mench, J. A. (2013). General principles for the welfare of animals in production systems: The underlying science and its application. The Veterinary Journal, 198(1), 19–27.
Fraser, D., Weary, D. M., Pajor, E. A., & Milligan, B. N. (1997). A scientific conception of animal welfare that reflects ethical concerns. Animal Welfare, 6, 187–205
von Keyserlingk, M. A. G., Rushen, J., de Passillé, A. M., & Weary, D. M.
(2009). Invited review: The welfare of dairy cattle—Key concepts and the role of science. Journal of Dairy Science, 92(9), 4101–4111. https://doi.org/10.3168/jds.2009-2326
Broom, D. M. (1986). Indicators of poor welfare. British Veterinary Journal,142(6), 524–526.
Caroline Ritter | Annabelle Beaver | Marina A. G. von Keyserlingk (2019) The complex relationship between welfare and reproduction in
cattle. Reprod Dom Anim. 2019;54(Suppl.3):29–37.
Lines 236-237 Please, consider to emphasize the relationship between welfare and reproduction parameters was assessed at herd and not at animal level.
Lines 240-241 did you mean "herd thinness, poor rumen fill and pregnancy rate"?
Line 246- replace cow by herd
Line 259 - herd's lameness
Line 351- "reproductive performance at herd level in this study"
Lines 352-355 suggestion: "The integrated approach on how some indicators of animal welfare might indirectly affect the reproductive performance of beef cattle in the extensive production system could be beneficial to address mitigation or intervention strategies for the improvement of reproductive efficiency in beef cattle."
Author Response
Please see attachment.
Thank you for the great suggestions!

Reviewer 2 Report
GENERAL COMMENTS
This is an interesting and novel study aiming to evaluate whether there is a relationship between several animal welfare indicators and the reproductive efficiency of beef cows in New Zealand managed in extensive pasture-based systems. There is few information regarding the relationships between animal welfare and productivity in beef cattle under pastoral (extensive) conditions, therefore it has merit to be published. Although the authors did not find important correlations between the animal welfare measures and reproductive and health measures, in my opinion this is due mainly to the low number of farms used, and particularly to the few records actually kept in each farm, and this deserves further discussion . For instance:
“The number of calves born was recorded at only 4/25 farms, tagging record was for 18/25 farms (at 2- 139 4 months) while 3/25 farms tagged calves only at weaning (mode 6 months), mortality only 4 farms…”. Hence there are only a few data on each farm, and moreover the percentages found for most variables in these farms are also low (for instance mortality). The sample size is not given for all variables.
I suggest to add a table (or add figures to Appendix A), where the reader can actually see how many measures there were for each variable. Moreover, some measures relate to the farm and other to individual animals (those measured directly by the authors on the animals during yarding). Sample size???
INTRODUCTION
L62 ..” the current paper aims to evaluate whether there is a relationship between the different animal welfare standards and the reproductive efficiency of beef cows in New Zealand managed in extensive pasture-based systems”.
The aim stated is broader than what was actually done: the authors did not use different AW standards…. they used the Welfare Assessment Protocol (WAP) developed by 72 Kaurivi or just several AW indicators (or some AW indicators) . Please rewrite considering this.
MATERIAL AND METHODS
I suggest to add information regarding the beef breeds present and some general environmental characteristics of the farms (climate, geographical location ) that could affect the welfare of the cattle on pasture: water availability, protection from rain/sun, some environmental measures (t°, rainfall, others). This would add to a general description of the conditions where the animals are kept and would help understanding the results. For instance dirtiness can be a good indicator of discomfort if the weather is wet, or the pasture lush, but not if it is hot and dry; they could suffer from heat stress but there is no indicator for this (perhaps the climatic conditions of the farm make it unnecessary, but the reader does not know the general characteristics).
In L88-89 it is explained that disbudding, castration, vaccination, diseases history, cattle deaths, access and type of water supply, feed/pasture condition, wintering practice, however the latter 3 were not considered or discussed here….?,
RESULTS
L166 Most farmers (11/25) did daily health checks on pregnant cows, whist 9/25 inspected cows once or twice a week…please correct whilst (comment: 11/25 is not really MOST FARMERS?)
L167 ….and others 5/25 do not inspect within a week. Did not
DISCUSSION
L224 …..in the present study because of incomplete record keeping…
Again, I think this is the main problem of this study, the few records for many of the important indicators that should be kept by farmers. I miss some more discussion on this, and how record keeping can improve efficiency (and probably also welfare) of the animals and of the farm. I think that much of the results and lack of correlations are due to the scarce numbers (although not shown) for many indicators….and also due to the low percentages of the indicators of poor welfare found (like low mortality rates, low % of thinness, lameness and health problems). Hence there were too few numbers for each indicator and the general status of the cows was not sufficiently compromised to find a negative impact on productive indicators. Should be further discussed.
L271 Interestingly, the frequency of yarding was related to the incidence of fearful behaviour (i.e. cattle attempting to escape the race or climbing/pushing on others), as also previously recorded [41, 42]…and …275…However, it appears that, in the present study, the commonness of fearful/agitated behaviour was just a temporary indication of the lack of familiarity of extensively managed cattle with yarding and handling, and not necessarily a permanent situation.
Regarding both previous statements it is not clear to me if the authors found that cows that were yarded more times per year were more fearful? This probably means that the handling/infrastructure is stressfull…??? Regarding stockpersonship, it would have been interesting to measure flight zone…or some other indicators on the pasture as well (positive/negative behaviours) . Please add some comments on this
L286 There was, however, a paradoxical relationship between welfare measures and weaning rate in the present study, inasmuch as weaning rate positively related to an increase in thinness, poor rumen fill and blindness in cows, but negatively to health checks. Most other reports e.g. [46-48], show the opposite….
Again, in my opinion most unexpected (or paradoxical) results are due to the low number of observations in many of the variables studied (due to not keeping records) and the discussion should center on this, enhancing the importance of keeping records in order to make improvements…On the whole, could it be concluded also that the welfare status of the cows was good considering the observations made directly by the authors?
Author Response
Please see the attachment.
Thank you very much for your insightful suggestions and comments.

Reviewer 3 Report
Overall a very interesting article, which takes up a current issue.
The manuscript is well written; the introduction gives sufficient background to the reader to understand the importance of the research. Methods and results are well described and discussion is sufficiently supported by the literature. Although the study has yielded contrasting results the authors discussed the results properly.
Some minor comments:
Keywords
Please add extensive systems
Introduction:
Line 54: please read: https://doi.org/10.3390/ani10112096 this paper may enrich the introduction
Line 263- 265: “Taken together, these results suggest that, in herds with low levels of lameness, long mating periods indicate low pressure for pregnancy; whereas in herds with high levels of lameness, long mating periods reflect a response to try and get as many cows pregnant as possible.” please rephrase to better understanding.
Appendix A: Farm F11 shows very critical measures that could affect the results, I’d try to remove it
Author Response
Please see the attachment.
Thank you very much for your valuable suggestions!

Round 2
Reviewer 1 Report
Congratulations!
Author Response
No comments or edits specified. So, no attachment necessary.
Reviewer 2 Report
Details only:
L276....and vice versa.. (delete one full stop)
L307.....in the cows,. (delete the comma)
L379-381: please revise sentence....the words "in this study" " and the study" are repeated
Please check title of Appendix A :........co-calf farms (cow-calf farms)
Author Response
Rectified the minor changes. Please see the attachment.
